# A Strong Baseline for Domain Adaptation and Generalization in Medical Imaging

**Li Yao**                                                        LI@ENLITIC.COM
**Jordan Prosky**                                            PROSKY@ENLITIC.COM
**Ben Covington**                                              BEN@ENLITIC.COM
**Kevin Lyman**                                              KEVIN@ENLITIC.COM

## Abstract

This work provides a strong baseline for the problem of multi-source multi-target domain adaptation and generalization in medical imaging. Using a diverse collection of ten chest X-ray datasets, we empirically demonstrate the benefits of training medical imaging deep learning models on varied patient populations for generalization to out-of-sample domains.

## 1. Introduction

Recent advancement in machine learning has created a surge in developing neural-network-based computer-aided diagnostic algorithms (Mazurowski et al., 2018). Despite widely acclaimed performance accompanied by rigorous regulatory assessments, most models rarely make their way into real world clinical environments. One major barrier that stops the successful technology transfer is that models do not generalize well to diverse patient populations. The situation is further aggravated by differences in acquisition parameters and manufacturing standards of medical devices. It is therefore not surprising that models trained on data from particular institutions perform well on in-domain validation sets while inevitably suffer from performance degradation when used in other domains. We refer to this as the problem of domain over-fitting.

Machine learning research has produced strategies and algorithms to mitigate domain over-fitting with the study of domain adaptation (DA) (Wang and Deng, 2018) and domain generalization (DG) (Li et al., 2017). Modeling in medical imaging, however, comes with unique challenges that are not faced in day-to-day computer vision tasks. For instance, medical images are typically of much higher resolution in 2D (and often are 3D or 4D), contain subtle artifacts, and have small regions of interest. Moreover, the interpretation of medical images can involve a high degree of uncertainty, even for highly-trained radiologists.

Reliable predictive modeling in medical imaging calls for remedies from DA and DG. This work illustrates the problem of domain over-fitting in the context of classifying chest X-rays (CXRs), the most commonly prescribed imaging exams worldwide. Experimental data are gathered from ten domains varied by their patient distributions, clinical environments, and global locations. We empirically show the phenomenon of performance degradation with inter-domain generalization. In this preliminary work, we suggest a simple solution which quantitatively shows its promise as a strong baseline for better generalization.

**Related work.** High performance in classification, detection and segmentation is regularly observed in retrospective clinical studies and publications. For instance, an AUC of 0.99 was reported in Lakhani and Sundaram (2017) for classifying pulmonary tuberculosis from CXRs. An average AUC of 0.96 was recorded in Dunnmon et al. (2018) in triaging normal and abnormal CXRs. A DICE score of 0.98 was shown in Weston et al. (2018) in segmenting body parts in abdominal CTs. A sensitivity of 0.96 was shown in Thian et al. (2019) in detecting fracture in wrist X-rays. Ueda et al. (2018) reported a sensitivity of 0.93 in detecting cerebral aneurysms in head MR angiography. As Kim et al. (2019) pointed out, however, most clinical publications do not contain a sufficient amount of external validation that is beyond the source domain, whose data is used to train the models. Among those that did, the recent work of Zech et al. (2018) empirically showed drastic performance gaps of models across three medical institutions in classifying pneumonia in CXRs. The work of Prevedello et al. (2019) also discussed a similar issue of coping with data heterogeneity, but offered no practical recommendations. Unlike previous work, we conduct an unprecedented study with ten datasets collected internationally, measuring the ability of state-of-the-art machine learning models to perform domain adaptation and domain generalization in the context of medical imaging. We establish baseline solutions that are intuitive and practical, and lead to better generalization performance in experiments.

## 2. Experiments

**Data.** We utilize ten datasets from diverse sources to empirically show the benefits of training with data from multiple domains for model generalization. For training, we use four publicly available datasets: ChestX-ray14 (Wang et al., 2017) (NIH), CheXpert (Irvin et al., 2019) (CHX), PadChest (Bustos et al., 2019) (PAD), Mimic-CXR (Johnson et al., 2019) (MIM), and one private data set from Australia (AUS). In addition, to evaluate generalization, we use one public data set, Open-i (Demner-Fushman et al., 2015) (OPI), and four private sets - one from Canada (CAN) and three from different sources in China (CHN$_1$, CHN$_2$, CHN$_3$). Table 1 below summarizes the data used in our experiments.

Table 1: Summary of our CXR data. We use a random 80/20 patient split when applicable.

| Dataset | Origin | # Patients | # Train Scans | # Test Scans |
|---------|--------|-----------|--------------|-------------|
| NIH | Bethesda, MD, USA | 30,806 | 89,322 | 22,798 |
| CHX | Stanford, CA, USA | 64,534 | 152,938 | 38,089 |
| PAD | Alicante, Spain | 67,216 | 88,207 | 22,347 |
| MIM | Boston, MA, USA | 62,592 | 200,874 | 49,170 |
| AUS | Australia | 125,000 | 100,000 | 25,000 |
| OPI | Bloomington, IN, USA | 3,670 | - | 3,670 |
| CAN | Canada | 18,000 | - | 19,000 |
| CHN$_1$ | China | 7,000 | - | 7,000 |
| CHN$_2$ | China | 3,000 | - | 3,000 |
| CHN$_3$ | China | 2,500 | - | 2,500 |

For all of the following experiments, we use a DenseNet-121 pretrained on ImageNet, and we are concerned with classifying CXRs as normal or abnormal. In the first set of

experiments, we train a model on each of the five training datasets and test on all ten sets. Table 2 shows AUCs from training on each source domain and evaluating on all target domains. We notice a couple of nice effects of training on all source domains. First, when aggregating all source domain data for training, test performance on those domains is essentially as good as training on any single source. Moreover, training on all sources simultaneously results in consistent improvement and yields the best performance on each of the five target domains, on which models are never trained on.

Table 2: AUCs on target domains when trained on different source domains.

|  |  | Source Domain | | | | | |
|---|---|---|---|---|---|---|---|
|  |  | NIH | CHX | PAD | MIM | AUS | ALL 5 |
| **Target Domain** | NIH | **0.769** | 0.732 | 0.751 | 0.756 | 0.744 | **0.769** |
|  | CHX | 0.823 | **0.866** | 0.816 | 0.851 | 0.782 | **0.862** |
|  | PAD | 0.811 | 0.807 | **0.853** | 0.803 | 0.832 | **0.850** |
|  | MIM | 0.814 | 0.825 | 0.793 | **0.854** | 0.783 | **0.853** |
|  | AUS | 0.795 | 0.776 | 0.808 | 0.769 | **0.848** | **0.841** |
|  | OPI | 0.758 | 0.744 | 0.783 | 0.723 | **0.791** | **0.786** |
|  | CAN | 0.772 | **0.789** | 0.783 | 0.783 | **0.787** | **0.788** |
|  | $CHN_1$ | 0.749 | 0.744 | 0.773 | 0.754 | 0.781 | **0.835** |
|  | $CHN_2$ | 0.771 | 0.725 | 0.770 | 0.716 | 0.786 | **0.805** |
|  | $CHN_3$ | 0.736 | 0.694 | 0.762 | 0.710 | **0.772** | **0.772** |

Figure 1 shows AUCs for experiments where one of the five source domains is left out during training. We observe that for NIH and CHX, leaving them out has a negligible impact on the model's performance. For AUS, PAD, and MIM, however, we notice what we expect: a moderate decrease in performance when the source is held out during training. There are many possible reasons to explain why performance is hurt more for some domains than for others, which we leave for further research.

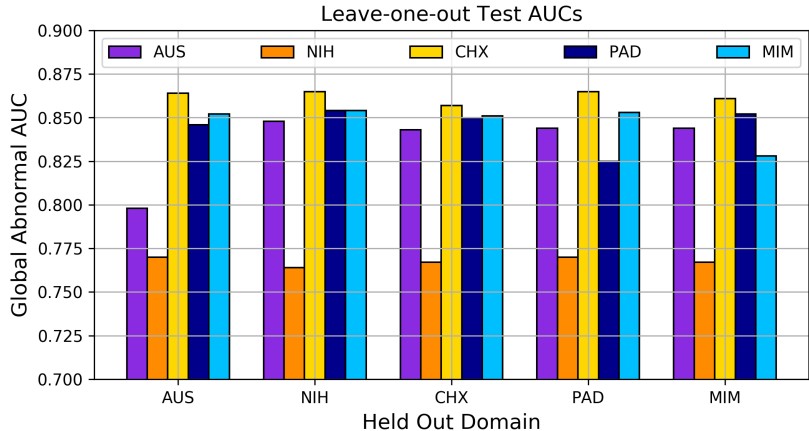

Figure 1: Varied performance of DG with leave-one-domain-out training.

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
