# OpenReview forum: "A Strong Baseline for Domain Adaptation and Generalization in Medical Imaging"
_MIDL.io/2019/Conference/Abstract — MIDL Abstract 2019_

### Official Review · AnonReviewer1 · 2019-04-29
**Simple but interesting validation study on a large amount of images**

**Rating:** 3
**Confidence:** 3

**Review:**

The authors train a standard architecture (Densenet-121) for classifying PA chest radiographs as normal or abnormal. They use 5 training datasets, a combination of the 5 sets, and test on 20% hold-out data from these 5 sets, and on test data from 5 additional sets. This makes for a large amount of data and this makes this study interesting. They demonstrate that training on the combination of the 5 sets works best, which is not very surprising but indeed provides a baseline. It would be nice if a few more details were provided on how the images were preprocessed (resolution, scaling of intensity values?) as this can have a big effect when testing on out-of-domain data. The related work section can be shortened by removing parts on non CXR work (if you want to review in general, much more space would be needed).

Could the authors comment on the reported performance of the computer systems versus human performance? I expect one can find in the literature indications that human performance on these datasets for this task is much higher.

The table with results would benefit from the addition of confidence intervals, or at least an indication of what the confidence intervals are.

---

### Official Review · AnonReviewer2 · 2019-05-02
**Very Useful**

**Rating:** 4
**Confidence:** 2

**Review:**

The authors create a baseline for domain adaptation and generalization methods by utilizing 10 different datasets of chest X-rays.

For experiments, a DenseNet-121 architecture pretrained on ImageNet is used for classifying the x-rays as normal or abnormal. AUCs are reported for all 10 target domains using different source domains for training. Training domains are 5 of the individual datasets, a union of the 5 datasets and a union of each 4 of them (one of the five datasets is left out to create 5 different leave-one-out sources).

I think the contribution of the paper is quite important. How well models trained on one dataset generalize to other datasets is a very important topic in medical imaging and the paper creates a useful baseline that will facilitate research in this area.

Quality: 5/5
Clarity: 5/5
Originality: 3/5
Significance: 5/5

---

### Decision · Program_Chairs · 2019-05-06
**Acceptance Decision**

Accept